# Gastric Bypass Associated Hyperammonemia (GaBHA): A Case Study, Scoping Review of the Literature, and Proposed New Pathophysiologic Mechanism

**DOI:** 10.3390/metabo15090573

**Published:** 2025-08-27

**Authors:** Andrew Z. Fenves, Dilara Hatipoglu, John C. Robinson, Michael M. Rothkopf

**Affiliations:** 1Nephrology Division, Massachusetts General Hospital, Boston, MA 02114, USA; 2Department of Medicine, Harvard Medical School, Boston, MA 02115, USA; 3Department of Gastroenterology and Hepatology, University of Pennsylvania, Philadelphia, PA 19104, USA; dilara.hatipoglu@pennmedicine.upenn.edu; 4Department of Pharmacy, Mayo Clinic, Phoenix, AZ 85054, USA; robinson.john@mayo.edu; 5Department of Medicine, Rutgers New Jersey Medical School, Newark, NJ 07101, USA; michael.rothkopf@rwjbh.org

**Keywords:** gastric bypass, encephalopathy, hyperammonemia, GaBHA syndrome

## Abstract

**Background/Objectives:** GaBHA syndrome (gastric bypass hyperammonemia) is an emerging new syndrome primarily in women who had prior Roux-en-Y gastric bypass surgery (RYGB) and then developed non-cirrhotic hyperammonemia with a high case–fatality ratio. Genetic and nutritional deficiencies have been implicated in the pathogenesis of this clinical condition, but none has been proven. We present an illustrative case and do a scoping review of the current literature in 58 patients with this diagnosis. **Methods:** A retrospective scoping literature review was utilized to identify patients who fulfilled 1. RYGB surgery, and 2. Hyperammonemic encephalopathy following the PRISMA extended checklist. We searched PubMed, MedLine, SCOPUS, and Web of Science databases. **Results:** We described the classic presenting symptoms and laboratory findings of this syndrome. We confirmed the female predominance (93%) and the high case–fatality ratio (32%). We then presented a novel hypothesis contending that arginine deficiency ultimately leads to a functional deficiency of the ornithine transcarbamolyase (OTC) enzyme, leading to the non-cirrhotic life-threatening hyperammonemia. Our hypothesis may also explain the high incidence of hypoglycemia found in these patients as we found in our search. Our proposed hypothesis may also be relevant to the occurrence of hyperammonemia in some solid organ transplant recipients. **Conclusions:** GaBHA syndrome is emerging as an important potential adverse outcome after RYGB surgery. It has a female predominance and a high case–fatality ratio. Arginine deficiency may explain the emergence of a functional OTC deficiency, which then leads to the severe hyperammonemia, and may also explain the frequent occurrence of hypo-glycemia in these patients.

## 1. Introduction

Roux-En-Y Gastric Bypass (RYGB) is an effective surgical treatment for morbid obesity but is associated with serious complications, including non-cirrhotic hyperammonemia [1,2] The syndrome was first described in 2008 in a series of five women who developed fatal, non-cirrhotic, hyperammonemic encephalopathy with no apparent etiology other than a prior history of RYGB surgery [3]. In 2015, another publication called attention to this syndrome, described 20 patients, and named it “gastric bypass” related hyperammonemia (GaBHA) syndrome [4]. The authors described the biochemical profile of these patients, noting nutritional deficiencies with no apparent cirrhosis and profoundly altered mental status. Patients also exhibited features that resembled urea cycle disorders, but no deleterious mutations of the *OTC* gene or copy number analysis have been identified [4]. A female predominance was noted.

Hyperammonemia can be life-threatening or result in serious neurological symptoms if left untreated and GaBHA carries a high mortality (up to 50%) [4,5]. Despite its severity, the management of GaBHA remains challenging, both because of the paucity of data about risk factors in the literature and a wide heterogeneity of treatment guidelines. Further, some cases of hyperammonemic encephalopathy in patients who had RYGB have likely not been classified as GaBHA, including one case we identified from literature published in 2020 [6].

The present paper has two purposes: first, to describe a case of GaBHA in the setting of acute respiratory distress syndrome (ARDS) secondary to COVID-19 with a special focus on the nutritional deficiencies and in particular low serum arginine level, and second, to perform an updated scoping review of reported cases to explore a novel mechanism that may explain the pathophysiology of this syndrome with potentially far-reaching consequences.

## 2. Methods

A retrospective scoping literature review was used to identify patients who fulfilled (1) RYGB surgery, and (2) hyperammonemia encephalopathy following the PRISMA extended checklist (Appendix A) [7] Exclusion criteria included patients with cirrhosis or evidence of liver disease (i.e., steatosis) without evaluation for cirrhosis (i.e., liver biopsy, imaging evidence of portal hypertension, or evaluation of synthetic dysfunction). The review was registered on the Open Science Framework (OSF), available online: https://doi.org/10.17605/OSF.IO/6HT5M (accessed on 23 July 2025). We searched PubMed, MedLine, SCOPUS, and Web of Science databases using the following terms: (“hyperammonemic encephalopathy” or “hyperammonemia” or “hyperammonemic”) and (“Roux-en-Y gastric bypass” or “Roux-en-Y” or “RYGB” or “Gastric Bypass” or “Bariatric Surgery” or “GaBHA”) to identify all potential cases of GaBHA. Conference abstracts were included in the inclusion of titles and abstracts. We did not impose any search filters when using the database. A total of 152 citations from all databases were uploaded into the Covidence systematic review software (Veritas Health Innovation, available at www.covidence.org), which immediately removed 79 exact duplicates. After applying the exclusion protocol to the screening of titles, abstracts, and full texts written by one author, a final set of 53 studies was included for the review. A summary of the PRISMA flow diagram is available in Figure 1. In addition to the 53 studies included in our scoping review, we identified 8 additional cases that were identified through available studies. These additional cases were included in our analysis if the primary source was accessible and provided sufficient clinical detail for inclusion. Variables extracted are summarized in Table 1. All calculations were performed using STATA/BE 18.0 (College Station, TX, USA). We adhered to all PRISMA guidelines when performing this review.

## 3. Case Presentation

A 57-year-old woman presented to the emergency department with three days of altered mental status. Fifteen days prior to presentation, the patient was hospitalized for acute deep vein thrombosis with suspected pulmonary embolism and a urinary tract infection (UTI). She was discharged home on apixaban and cephalexin. Her past medical history was notable for obesity with a successful RYGB 30 years earlier.

Baseline laboratory results revealed a selenium level of 30 ng/mL (reference, 70–150 ng/mL); a zinc level of 26 mcg/dL (reference, 66–110 mcg/dL); a glutamate level of 1078 nmol/L (reference 371–957); and a urine orotic acid level of 2.7 mmol/mol Cr (reference, 0.4–1.2 mmol/mol Cr). A full amino acid panel is presented in Table 1. Note the low arginine level of 31.

In the emergency department, the patient was febrile (38.5 °C) and hypoxemic requiring 3 L of oxygen via nasal cannula. She was altered with an inability to follow commands. The patient tested positive for COVID-19 with a chest radiograph demonstrating patchy interstitial infiltrates with concern for ARDS. A chest computed tomography (CT) with pulmonary artery angiography was positive for acute bilateral pulmonary embolisms. There was no laboratory or radiographic evidence of cirrhosis. CT head was negative for acute pathology. The patient was initiated on enoxaparin for the pulmonary emboli, dexamethasone and remdesivir for COVID-19, and ceftriaxone for her UTI.

On day 2 of her hospitalization, urine culture yielded extended spectrum beta-lactamase (ESBL) Klebsiella pneumoniae, and antibiotic treatment was escalated to ertapenem. Laboratory results were significant for an elevated serum ammonia level of 59 mmol/L, an aspartate aminotransferase of 74 U/L, and an alanine aminotransferase of 87 U/L. On day 4, the patient followed simple commands despite an increase in serum ammonia to 74 mmol/L. An enteral tube was placed to provide nutrition. Within 48 h of starting enteral nutrition, her serum ammonia increased to 120 mmol/L and she required intubation for airway protection.

That day blood samples were sent for serum selenium, zinc, and an amino acid panel, and a urine orotic acid level was obtained. Treatment for a suspected diagnosis of GaBHA syndrome was initiated. Enteral nutrition was stopped to restrict protein, and the patient was given lactulose, rifaximin, and a daily solution of 5% dextrose that contained vitamins, trace elements, and additional zinc supplementation. Within 72 h, the serum ammonia was 16 mmol/L and the patient was following commands. Enteral nutrition was gradually advanced to goal over five days while monitoring serum ammonia, which remained normal. During her ICU stay, she was treated for a ventilator-associated pneumonia, and she had a spontaneous retroperitoneal bleed that required embolization of the superior gluteal artery. The patient spent 10 days on the ventilator, 14 days in the intensive care unit, and 28 days in the hospital overall before being discharged home with outpatient follow up.

## 4. Review of Cases in the Literature

Table 2 summarizes 58 cases of GaBHA encountered in the literature from 2013 to 2024. Case patients averaged 45.4 years of age (SD = 9.9; n = 57) and were 8.4 years post-RYGB (SD = 6.9; n = 47). Of reported cases, 93.1% (n = 54/58) occurred in female patients, consistent with previous literature. The majority, 62% (n = 36/58) of patients, survived.

The average elevated peak ammonia level reported was 229 μmol/L (SD = 150; range 40.1 to 903; n = 55). Urine orotic acid levels were reported in 36% of cases (n = 21). A numeric value for urine orotic acid was reported in 12 patients (mean 2.12 mmol/mol, SD 0.79); 7 cases were reported as normal and 1 case as “high”. Glutamine levels were reported in 35% of cases (n = 11). Glutamine was elevated in reported in 25 cases. The average glutamine level was 1272.9 μmol/L (17 cases); 7 cases were reported as normal and 1 case as “high”. Citrulline levels were reported in 29% of cases (n = 9). Citrulline was reported in 21 cases; the average citrulline level was 17.2 Umol/L (SD 10.9, n = 18), with 2 reported as normal and 1 reported as “low”. Albumin was reported in 42 cases, and the average was 1.8 (SD = 0.54, n = 36), with reported 4 as “normal” and 2 as “low”. Arginine levels were reported in 14 cases. The mean value was 46.2 μmol/L (SD 38.2, n = 10); 1 patient had a reported normal value, while three patients had a “low” value. Zinc levels were obtained in 34 cases with an average value of 34.4 (SD 16.6, n = 24); nine cases reported as normal and one as “low”. Thirteen of the cases had genetic screening for urea cycle disorder. Among the 22 patients who mentioned glucose trends, 68% (15) demonstrated concern for hypoglycemia.

Treatment with lactulose and rifaximin continues to remain the standard of care. Other treatments include ammonia scavengers (sodium benzoate and sodium phenylacetate) and nutritional and amino acid supplements, commonly including zinc, pyridoxine, thiamine, intravenous arginine, multivitamins, and L-carnitine. There were at least eighteen instances of renal replacement therapy.

## 5. Discussion

We performed a comprehensive literature review of described cases of GaBHA along with the identification of a recent case in the setting of COVID ARDS. Our findings are consistent with previous studies that have reported an increased risk of hyperammonemia after gastric bypass surgery, particularly in patients with mild liver disease (such as fatty liver disease or early-stage fibrosis) or other underlying conditions that may affect ammonia metabolism. We noticed that in the case presented here, the patient had low serum arginine levels. This has stimulated us to propose a potential new mechanism for the development of hyperammonemia in these patients with otherwise relatively normal liver function.

The early suspicion for an etiology fell on the urea cycle, especially because of the female predominance. This suggested that perhaps our patients had an OTC deficiency. Since this is an X-linked gene, we theorized that the affected women were heterozygotes for OTC deficiency and hence were normally functioning until after the RYGB surgery. However, this hypothesis failed when every woman tested normal for wild type genes of OTC.

Still, we did find that the organic amino acid profiles of GaBHA patients fit an OTC deficient state. Furthermore, when the actual OTC enzymatic activity was tested in vivo (in frozen liver tissue) in at least three cases, two of them had zero or very low activity. This led us to propose that these patients had acquired, not genetic, OTC dysfunction.

In exploring the possible causes for acquired OTC dysfunction, we noticed that several of the reported patients had low or low normal arginine levels. This led us to develop a novel hypothesis for the pathogenesis of GaBHA based on arginine deficiency.

We have support for this concept based on the available genetic studies, in vivo enzymatic activity, urine orotic acid levels, and plasma amino acid profiles. Although the data are far from complete, we have enough to structure a theory and support most of its tenants. If proven valid, this pathophysiology may help explain other causes for non-cirrhotic hyperammonemia, such as that seen after organ (especially lung) transplantation [51,52,53].

Any malfunction of OTC activity, whether due to the enzyme itself, required cofactors or required precursor substrates, would impair the urea cycle (Figure 2), leading to severe hyperammonemia. This would be particularly true for patients who were consuming high protein diets and/or protein supplements (a general recommendation for post-RYGB patients), which would increase the ammonia load significantly. Without normal OTC activity, ammonia has no entry point to the urea cycle. Under such circumstances, ammonia produced from the catabolism of amino acids would accumulate based on the protein load and protein catabolic rate.

Ammonia enters the urea cycle after it is incorporated into carbamoyl phosphate, an OTC substrate. OTC transfers the amino group from carbamoyl phosphate to ornithine to form citrulline (Figure 3). If OTC is either absent or dysfunctional, carbamoyl phosphate is not converted and citrulline is not produced. Excess carbamoyl phosphate is later converted to orotic acid. Therefore, OTC dysfunction increases ammonia and orotic acid levels and reduces circulating citrulline. This pattern was observed in our current patient and in many of those examined in our literature review. The pattern observed in GaBHA is identical to that seen in genetic OTC deficiency.

Given that many GaBHA patients have not been shown to have abnormalities of the *OTC* gene, it was logical to explore reasons for functional defects of OTC activity. OTC requires the presence of cofactors, including zinc and vitamin A. Zinc levels were low in the current GaBHA patient and in many of those reported in the literature. We do not have data on vitamin A levels, but vitamin A deficiency is commonly seen in RYGB patients. Therefore, OTC function could have been impaired by inadequate tissue cofactors.

OTC dysfunction could also be caused by a lack of its substrate, carbamoyl phosphate. OTC acts by transferring the carbamoyl group (NH3, C, O) of carbamoyl phosphate to ornithine. This converts bi-amino ornithine to tri-amino citrulline (Figure 3). The amino group added to form citrulline was derived from ammonia through the formation of carbamoyl phosphate (Figure 4). The body’s ability to dispose of ammonia is dependent upon this step.

The formation of carbamoyl phosphate occurs in three hepatic mitochondrial steps performed by the complex multidomain enzyme, carbamoyl phosphate synthetase I (CPS-I; Figure 4), which is the rate-limiting step of the urea cycle. In the first step of CPS-I activity, bicarbonate is phosphorylated to carboxyphosphate. The second CPS-I step involves the addition of NH3, which forms carbamic acid and releases phosphate. In the third step of CPS-I, carbamic acid is phosphorylated to form carbamoyl phosphate (CP). Both phosphorylation steps (1 and 3) require the allosteric activator, N-acetyl-L-glutamate (NAG). NAG levels operate CPS-I as a switch that is in the on position when the NAG levels are high, and in the off position when the NAG levels are low. In this way, NAG acts to prevent amino acid depletion through the urea cycle [55].

In the absence of NAG, CPS-I operates at less than 2% of its activity with NAG saturation [56]. The binding of NAG occurs at the C-terminal domain of CPS-I, which triggers conformational changes in the two phosphorylation domains of the enzyme. This results in a remodeling that stabilizes the conformation and allows for migration of carbamate to the second phosphorylation site (Figure 5).

Based on an understanding of the “on-off switching role” of NAG on CPS-I, we can trace the biochemistry back one more step, to NAG formation. NAG is formed by combining the amino acid glutamate with acetyl CoA under the control of NAG synthase (NAGS). Importantly, NAGS requires arginine for activation (Figure 6).

NAG deficiency is most often considered in relation to genetic diseases of the urea cycle. First recognized in 1981, genetic mutations of NAGS are the rarest of the urea cycle disorders. As in CPS-I deficiency, NAGS deficiency presents with elevated glutamine, reduced citrulline, and elevated ammonia levels [57] Similar to CPS-I deficiency, orotic acid levels are not elevated in NAGS deficiency because CPS-I activity (and production of CP) is completely shut down.

However, orotic acid levels were elevated in 66% of GaBHA patients tested for that metabolite. This indicates that CP was produced in excess in GaBHA patients, not processed by OTC and then converted to orotate. The mechanism for this observation is likely through a related pathway, CPS-II. This pathway involves the conversion of glutamine into CP and carbamoyl aspartate (CA) for de novo pyrimidine synthesis.

Glutamine levels were elevated in the current GaBHA case and in many of those reported in the literature. Glutamine levels are commonly elevated in the presence of hyperammonemia because of activity of the enzyme glutamine synthetase, which converts glutamic acid to glutamine. This pathway is an important alternative pathway for disposal of excess ammonia by CNS astrocytes [58].

Based on the biochemistry described above, it is feasible to assert that arginine deficiency leads to a cascade of events which ultimately diminish OTC activity. The lack of arginine causes a reduction in NAG production. Reduced NAG impedes activity of the rate-limiting urea cycle enzyme CPS-I, which reduces the production of carbamoyl phosphate. Limited access to carbamoyl phosphate halts the conversion of ornithine to citrulline by OTC and inactivates the urea cycle (Figure 7). This results in hyperammonemia and elevated serum glutamine. Excess glutamine is processed via CPS-II to form orotic acid.

## 6. Role of OTC Dysfunction in Fasting Hypoglycemia

We have observed that some patients with GaBHA also manifest fasting hypoglycemia. Reactive hypoglycemia has been reported in post-gastric bypass patients, possibly attributed to nesidioblastosis of pancreatic islet cells.

It is reasonable to consider that there could be a link to the urea cycle impairment as was outlined above. In the case of dysfunctional OTC activity and impairment of the urea cycle, decreased production of substances produced by steps after OTC would be expected. This includes fumarate production through the action of AS lyase.

Fumarate plays an important role in gluconeogenesis, particularly in patients with malabsorption and malnutrition. Therefore, it is possible that the same disorder of OTC function we have proposed for the production of hyperammonemia could also be responsible for the fasting hypoglycemia observed in patients with GaBHA (Figure 7) [59].

## 7. The Conditional Essentiality of Arginine in Stress Metabolism

Although arginine is one of the 20 “alphabet” amino acids necessary for protein synthesis and needed to form multiple compounds (i.e., nitric oxide, polyamines, proline, glutamate, creatine, etc.) it is not considered an essential dietary component [60] This is because the body has ample capacity to manufacture arginine through pathways in the liver, intestine, and kidneys [61]. However, when the need for arginine exceeds the body’s ability to produce it, dietary intake of arginine becomes critical. Under these circumstances, arginine is considered “conditionally essential” [62]. This situation commonly occurs in stress states such as critical illnesses [63]. Furthermore, many of these same conditions may cause the release of arginase, which can lead to the destruction of existing arginine reserves [64]. Thus, a critically ill patient could easily experience decreased arginine production, reduced nutritional arginine intake, and increased destruction of arginine stores. In combination, these processes could result in a severe lack of arginine availability for NAG production. As stated above, the absence of NAG would be expected to severely limit CPS-I activity, resulting in urea cycle dysfunction.

It is interesting to note that transplant patients also experience non-cirrhotic hyperammonemia [65,66,67]. This condition is especially significant in lung transplantation, where it confers a high mortality rate [52]. The etiology of hyperammonemia after transplant is poorly understood [68]. It is plausible that arginine deficiency in the critically ill transplant patient could have a similar role to producing hyperammonemia as we have hypothesized for GaBHA patients. Interestingly, a recent animal study showed high levels of arginase in pig donor lungs after normothermic ex vivo lung perfusion [53]. This suggests that critically ill lung transplant patients may have a higher risk of arginine depletion due to arginase activity in the donor lung(s).

One may ask when clinicians should suspect this clinical entity. We and others have observed the severe hypoalbuminemia in this syndrome. In addition, those patients who recover often normalize their serum albumin levels. Accordingly, we suggest serial serum albumin measurements in the RYGB patients and obtaining ammonia levels if the serum albumin falls significantly.

Strengths of this review include comprehensive data review from multiple different databases and publication types including conference abstracts. To our knowledge, this is the largest collection of reported potential cases of GaBHA to be published. The purpose of this review was to gather comprehensive data regarding published cases as well as propose a potential pathophysiologic mechanism based on this review. Limitations of the scoping review include variability in the clinical detail reported across studies as well as reliance on case reports or abstract data. We mitigated this by only including studies that were available for review and that demonstrated the minimum requirement in diagnostic workup outlined in the methodology.

## 8. Conclusions

We presented an additional case of GaBHA syndrome to an enlarging group of patients previously reported in the literature. We confirmed the female dominance and the high case mortality of this syndrome. We also confirmed the hypothesis that GaBHA syndrome is the result of effective OTC deficiency, and the fact that, so far, the majority tested do not show genetic abnormality. Finally, we proposed a novel pathophysiologic mechanism for the GaBHA syndrome, namely arginine deficiency, which ultimately leads to OTC enzymatic failure, with resultant often fatal hyperammonemia. This may also explain the often-observed hypoglycemia. In addition, this mechanism may also be active in hyperammonemia associated with certain solid organ transplants, especially the lung.

## Figures and Tables

**Figure 1 metabolites-15-00573-f001:**
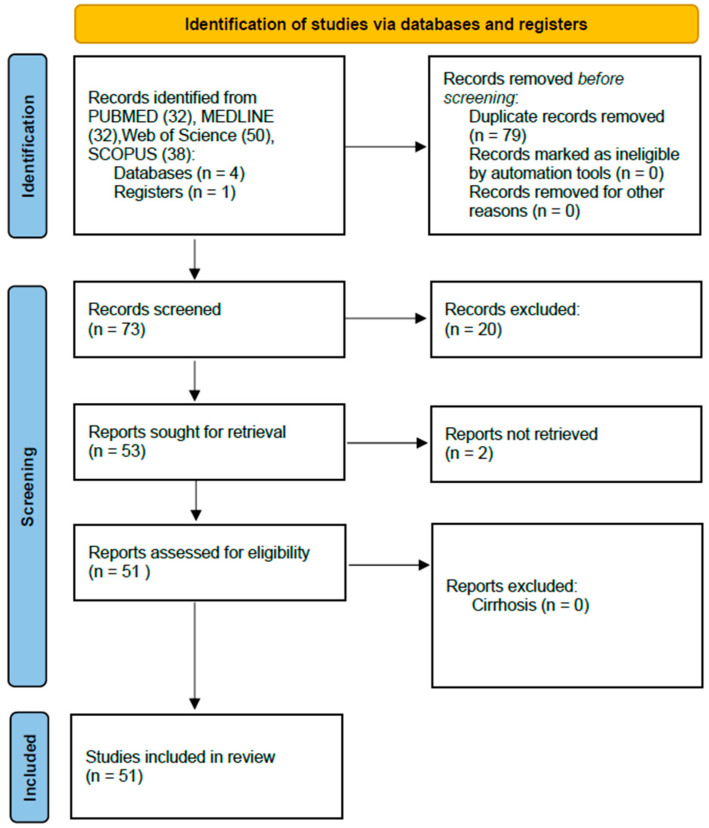
PRISMA 2020 flow diagram for scoping review. A search was performed in the PubMed, MedLine, SCOPUS, and Web of Science databases using the following terms: (“hyperammonemic encephalopathy” or “hyperammonemia” or “hyperammonemic”) and (“Roux-en-Y gastric bypass” or “Roux-en-Y” or “RYGB” or “Gastric Bypass” or “Bariatric Surgery” or “GaBHA”) to identify all potential cases of GaBHA. Exclusion criteria included patients with cirrhosis and those with evidence of liver disease but no evaluation for cirrhosis.

**Figure 2 metabolites-15-00573-f002:**
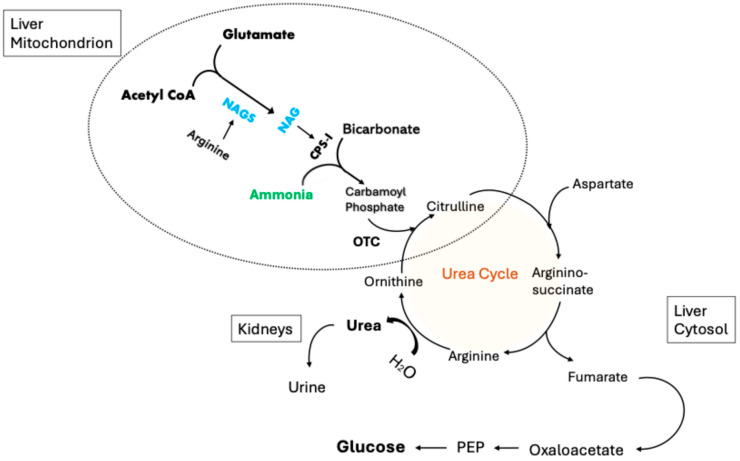
The urea cycle in hepatic mitochondria and cytosol. N-acetyl glutamate (NAG) is produced from glutamate and acetyl CoA by NAG synthase. NAG stimulates mitochondrial CPS-I to condense ammonia with bicarbonate, generating carbamoyl phosphate (CP). CP is used by OTC to transfer ammonia’s amino group to ornithine, forming citrulline. Citrulline leaves the mitochondria where it is combined with aspartate to form arginosuccinate (AS) by AS synthetase. AS is condensed into fumarate and arginine by AS lyase. The enzyme arginase hydrolyzes arginine into urea and ornithine.

**Figure 3 metabolites-15-00573-f003:**
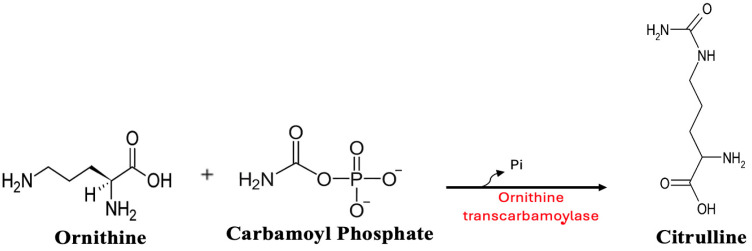
Ornithine acquires the carbamoyl group of carbamoyl phosphate in a reaction catalyzed by ornithine transcarbamoylase (OTC) to form citrulline [54].

**Figure 4 metabolites-15-00573-f004:**
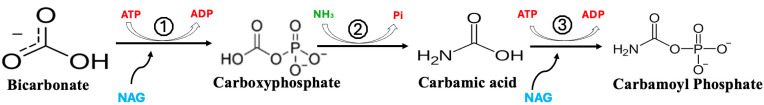
Carbamoyl phosphate is formed in three steps by the hepatic mitochondrial enzyme complex, carbamoyl phosphate synthetase I (CPS-I). ① Bicarbonate is phosphorylated to carboxyphosphate. ② The addition of NH3 condenses carboxyphosphate to form carbamic acid and releases phosphate. ③ Carbamic acid is phosphorylated to form carbamoyl phosphate (CP). The consumption of two molecules of ATP makes this pathway irreversible. Both phosphorylation steps (1 and 3) require the allosteric regulator *N*-acetylglutamate (NAG) [54].

**Figure 5 metabolites-15-00573-f005:**
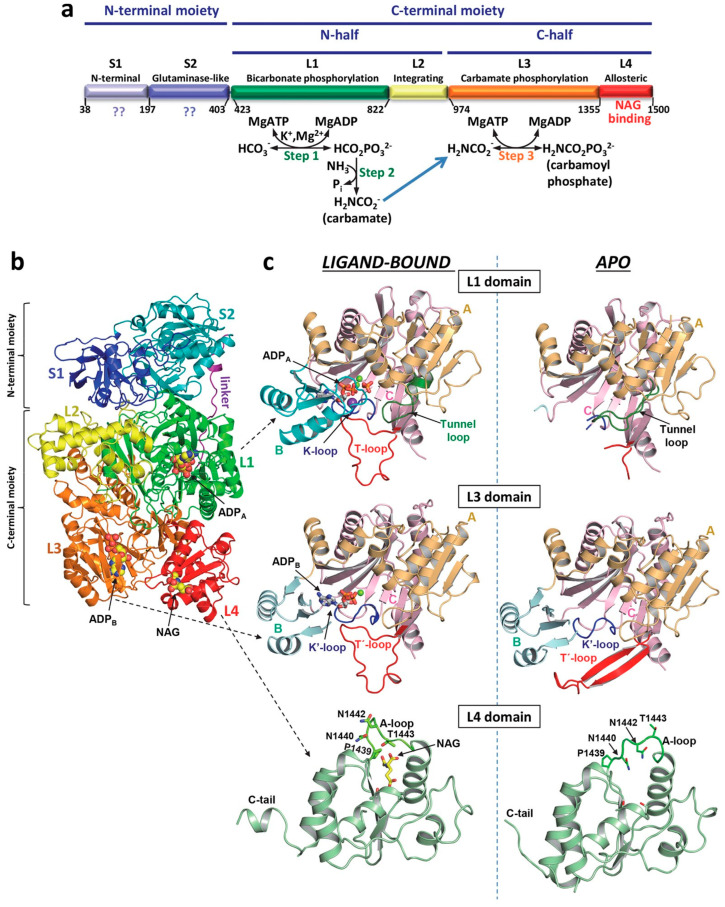
Structure of human CPS-I. (**a**) Scheme of the mature CPS-I polypeptide, indicating its two moieties (top), which are homologous to the small and large subunits of CPS from *E. coli*. Also indicated are the two sequence-related halves of the large moiety (middle). Domain composition with names and boundaries, given as residue numbers, is indicated in the lower bar. The three steps of the CPS-I reaction are shown below the corresponding domains where they occur. The blue arrow indicates carbamate migration between the phosphorylation active centers, a process contributed mainly by residues from both phosphorylation domains. (**b**) Cartoon representation of the CPS-I monomer, with domains shown in different colors and labeled. The structure depicted corresponds to the ligand-bound conformation. The two ADPs and NAG molecules found bound to CPS-I are shown in space-filling representation. In (**c**), domains L1, L3 and L4 are shown expanded, in the ligand-bound (left) and apo (right) forms. In L1 and L3 the three subdomains A, B, and C, and important loops mentioned in the text are labeled and represented with different colors. In the L4 domain, some residues from the A-loop are shown (as sticks) to highlight the large structural changes of these side chains when NAG is bound. The ADP molecules, a phosphate, and NAG are shown in rods representation, magnesium ions are shown as green spheres, and the potassium ion bound to the K-loop is shown as a violet sphere. From de Cima, S. et al. [55]. Creative Commons Attribution 4.0 International License.

**Figure 6 metabolites-15-00573-f006:**
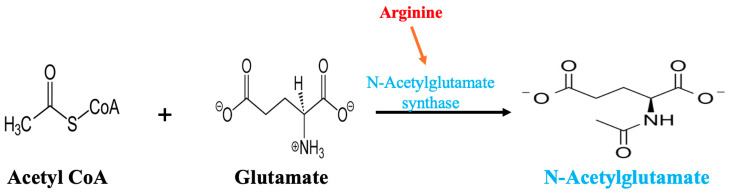
The allosteric regulator *N*-acetylglutamate (NAG) is required for carbamoyl phosphate synthetase (CPS-I) activity. NAG is synthesized by *N*-acetylglutamate synthase (NAGS). But the NAGS enzyme is itself activated by arginine and suppressed when arginine is absent [54].

**Figure 7 metabolites-15-00573-f007:**
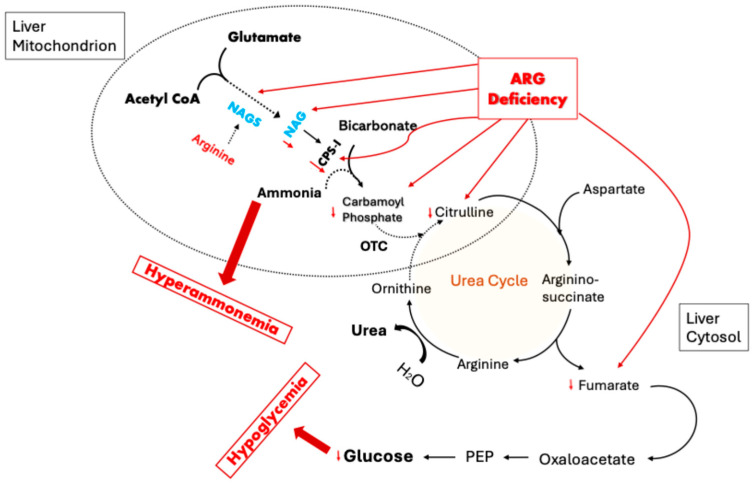
The impact of arginine deficiency on OTC precursors and downstream products. Arginine deficiency would be expected to reduce NAGS activity, leading to decreased intramitochodrial NAG. Decreased NAG would likely decrease CPS-I activity, increasing ammonia levels and reducing CP. Reduced CP availability would decrease Citrulline, AS, and fumarate levels. Reduced fumarate would decrease the body’s capacity to perform gluconeogenesis, favoring fasting hypoglycemia.

**Table 1 metabolites-15-00573-t001:** Nutritional and amino acid profile in a case of GaBHA.

Parameter	Result	Reference Range
Phosphoserine	0	<18
Phosphoethanolamine	<2	<12
Taurine	26 (L)	42–156
Asparagine	62	37–92
Serine	180	63–187
Hydroxyproline	37 (H)	4–29
Glycine	257	126–490
Glutamine	1078 (H)	371–957
Aspartic acid	5	<7
Ethanolamine	24	<67
Histidine	77	39–123
Threonine	162	85–231
Citrulline	21	17–46
Sarcosine	5 (H)	<5
Beta-Alanine	8	<29
Alanine	327	200–579
Glutamic acid	78	12–113
1-Methylhistidine	2	<28
3-Methylhistidine	3	2–9
Argininosuccinic acid	0	<2
Carnosine	0	<1
Anserine	0	<1
Homocitrulline	0	<2
Arginine	31 (L)	32–120
Alpha-aminoadipic acid	1	<3
Gamma-amino-n-butyric acid	0	<2
Beta-aminoisobutyric acid	7 (H)	<5
Alpha-amino-n-butyric acid	31	9–37
Hydroxylysine	1	<2
Proline	283	97–368
Ornithine	60	38–130
Cystathionine	<1	<5
Cystine	23	3–95
Lysine	149	103–255
Methionine	13	4–44
Valine	93 (L)	136–309
Tyrosine	42	31–90
Isoleucine	29 (L)	36–107
Leucine	37 (L)	68–183
Phenylalanine	52	35–80
Tryptophan	18 (L)	29–77
Allo-isoleucine	1	<5

Note: All the values are reported in nmol/mL.

**Table 2 metabolites-15-00573-t002:** Review of cases of GaBHA from 2007–2024.

Review Article	Age (y)	Sex	Years Since RYGB Surgery	Outcome	Peak Ammonia (μmol/L)	Urine Orotic Acid (mmol/mol)	Gln (μmol/L)	Cit (μmol/L)	Albumin (g/dL)	Zinc Level (mcg/dL)	Arginine Level (μmol/L)	Report of Hypoglycemia	OTC Testing	Treatment Received
Fenves [3,4]	50	F	1.4	Deceased	138				2			Yes		
	48	F	0.3	Deceased	50.7				2			Yes		
	26	F	1.4	Deceased	286.7				1.3			Yes		
	58	F	28	Deceased	44.8				1.5			Yes		
	41	F	6	Deceased	40.1				2.1			Yes		
	54	F		Deceased	205		1824		0.8	19		No		
	52	F	1	Survived	171		1004	33	1.1	58		Yes	15% activity of OTC in liver bx	
	49	F	11	Survived	96				1.5	23		Yes		
	46	F		Deceased	450	3.6			1.7	35		No		
	38	F	4.4	Survived	157	1.8	905		1.4	25		Yes		
	44	F	10	Deceased	258		1700		1.7	86		Yes		
	41	F	0.5	Survived	99	2.7	858	13	1.8	41		Yes		
	69	F	1.3	Survived	201	0.9	811	40	2.5	41			Normal activity of OTC in liver bx	
	37	F	3	Deceased	180				1.8	31		No		
Rashti [8]	43	F	10	Deceased	191					Normal				Lactulose, Rifaximin
Hahn [9]	41	F	11	Deceased	446		1998						Negative gene testing for OTC and CPS-1 mutation	Dialysis, ammonul, cyclinex-2 (aminoacid modified food), citrulline
Elhassan [10]	47	F	9	Survived	116					Normal			Negative screening for urea cycle disorders	Lactulose, rifaximin, pyridoxine, thiamine, zinc
Postigo Jasahu [11]	52	F		Survived		Normal			Normal	Normal				Lactulose, rifaximin, Golytely
Kromas [12]	56	M	2	Survived	91.5	1.4				Normal				Lactulose, sodium benzoate, protein restriction
Khanal [13]	45	F		Survived						Normal				CRRT
Acharya [14]	42	F	2	Deceased	491	2.3	1363	43	1.8	23	130	No		Lactulose, rifaximin, D5, VitK,
O’Donnell-Luria [15]	47	F	10	Survived	138	2.8	1391	17	1.9	Normal	20		Negative testing for OTC, GLUD1, and PCCA	Lactulose, rifaximin, added sodium phenylacetate and sodium benzoate, arginine, N-carbamoyl glutamate. IVF without protein and then TPN with slow protein advancement. Vitamin and minerals
Nagarur and Fenves [16]	42	F	11	Deceased	498	Normal	Normal	12	1.4	34				Lactulose, rifaximin, HDx1, Zinc, thiamine, Dextrose
Grogg [17]	45	F	15	Survived	82.6	2.4	Normal	9	Normal	Normal	24			Lactulose, rifaximin, low-protein parenteral diet
Ahmed [18]	50	F	10	Survived	.	Normal	Normal	Normal	Normal	Normal	normal			Lactulose
Borreggine [19]	44	F	11	Survived	186	1.9	Normal	Normal	2.2	29			Negative Sequencing analysis and deletion/duplication analysis of OTC deficiency gene	OVF, thiamine, selenium, zinc, then lactulose, rifaximin, and levocarnitine
Salcedo [20]	48	F	20	Survived	173				2.3	54		yes		Lactulose
Castineira [21]	42	F	1	Survived	190	1.63			1.7					Lactulose, sodium benzoate, sodium phenylacetate, rifaximin, zinc, arginine (po), IV lipids, and dextrose
Krishnan [22]	45	F	4	Survived	432		396 (mmol/L)	9	Normal	49	34			CRRT, supplemental nutrition fortifying her parenteral feeds with the essential amino acid combinations found deficient
Hendrikx/ Lloyd and Fenves [6,23]	28	F	7	Survived	251				1	45				Adjustment to enteral nutrition to slowly increase protein as IV glucose was tapered
Purpura [24]	57	F		Survived	251					14				CRRT, protein restriction, vitamin and L-carnitine supplementation
Vartanyan [25]	45	F		Survived	192					Normal				Lactulose, rifaximin, CRRT, arginine, sodium benzoate, zinc and copper supplementation
Bergagnini [26]	45	F	16	Survived	227		Normal							CVVHD, lactulose, rifaximin, low protein
Sun [27]	24	F	3	Survived	169				1.7					Initially, PN and albumin but stopped those and started duphalac enema, lactulose, probiotics, ornithine aspartate THEN gastrostomy and enteral nutrition
Patel [28]	43	F		Deceased	169									Lactulose, rifaximin, zinc, MTV, L-carnitine
Rosenberg [29]	58	F	20	Deceased	290					21.1			Genetic testing for urea cycle disorder negative	Lactulose, rifaximin, CRRT
Vinegard [30]	44	F	14	Survived	268.5		Normal	4	1.5					sodium benzoate, N-carbamylglutamate, l-arginine, carnitine, and low-protein TPN, Zinc, thiamine, Vit B, C, E; Hemofiltration
Sharma [31]	38	F	16	Survived	314				1.9	35.3				lactulose, albumin, thiamine, low protein, IV sodium benzoate, nutrient supplementation; HD
Robinson [32]	52	M	7	Deceased	176				1.9	21				lactulose, rifaximin, zinc, MTV with minerals
Dace [33]	49	F	14	Survived	234	Normal	2235	8	1.5	4.7				(BIMDG guidelines) sodium benzoate, L-arginine, rifaximin, lactulose, oral amino acid, vitamins supplementation, insulin, pancreatic enzymes
Kim [34]		F	13	Deceased	135				<1.5	29.7				Lactulose, enteral tube-feeding, antibiotics
Summar [35]	34	F	0.7	Deceased	442	Normal					Low		NAGs deficiency found on liver biopsy testing	Dialysis, phenylbutyrate, citrulline
Hu [36]	29	F	0.1	Survived	92	3.2	2018	17	2.1	30	102		OTC deficiency on biopsy of liver, <1% enzyme activity. Genetic testing was negative for OTC mutation	Lactulose, parenteral nutrition, carnitine supplementation,
Limketkai [37]	35	F	6	Survived	342			19	1.6		39	No		Thiamin, lactulose, neomycin, hemodialysis, oral levocarnitine
Estrella [38]	52	F	0.5	Survived	155	Normal	448	19	1.8		28	No	Negative CPS1 and NAGS gene sequencing	Oral metronidazole, cholestyramine, erthyromycin, sodium benzoate, arginine, parenteral nutrition, reversal of RYGB
Rogal [39]	58	M	7	Survived	306		High		3.8			No		Closure of splenorenal shunt
Singh [40]	39	F	4	Survived	477	1.4	644	15	2.8		16			Thiamine, lactulose, multivitamin, zinc, L-carnitine, rifaximin, IV arginine, dialysis
Loeffler [41]	41	F	16	Survived	200		808.7	7.3		38		Yes		Glucose, thiamine, lactulose, rifaximin, zinc
Zeghlache [42]	49	F	7	Survived	104	normal		13	1.9	38	42	Yes		Lactulose, rifaximin
Kjaergaard [43]	46	F	11	Survived	115		normal		1.7		low	Yes		Parenteral nutrition, vitamin K, lactulose
Bonasso [44]	17	F	0.05	Deceased	903							Yes		Hemodialysis
Thusay [45]	54	F	23	Deceased	168									CRRT
Sadlik [46]	44	F		Survived	305				low	low				CRRT, lactulose, rifaximin, carnitine, vitamin and nutrient supplementation, parenteral nutrition
Fanous [2]	65	F		Deceased	264	high	2254	low			low		Negative urea cycle disorder genetic panel	Lactulose, rifaximin, dialysis, levocarnitine, sodium phenylbutyrate, zinc
Bendrick [47]	56	M	19	Survived	123								Negative urea cycle disorder genetic panel	Low-protein diet, lactulose, TPN
Laoye [48]	42	F	6	Survived	192			18	2.4	44				Intravenous L-carnitine, hemodialysis, lactulose, rifaximin
Roth [49]	65	F		Survived	216	1.6	2254	14	2.7		27		Negative urea cycle disorder genetic panel	Lactulose, rifaximin, L-carnitine, zinc, multivitamin, IV sodium phenylacetate-sodium benzoate, arginine infusion
Lloyd [50]	35	F		Survived	100				1.79					Lactulose, rifaximin

## Data Availability

The raw data supporting the conclusions of this article will be made available by the authors on request.

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
