# Peer review of "Gastric Bypass Associated Hyperammonemia (GaBHA): A Case Study, Scoping Review of the Literature, and Proposed New Pathophysiologic Mechanism"

_metabolites, 2025, doi:10.3390/metabo15090573_

Round 1
Reviewer 1 Report
Comments and Suggestions for Authors
I find this a very interesting clinical case, due to the clinical implications of its early diagnosis and treatment.
General aspects:
I believe the discussion should focus more on the clinical aspects of the disease and narrow the scope of the biochemical characteristics of the metabolism of this urea cycle disorder (we suggest including it as supplementary material).
However, it is striking that no cases have been described in malabsorptive surgeries (such as duodenal switches). It is also important to understand the involvement of the microbiota, bacterial overgrowth, or exocrine pancreatic insufficiency in the acquired development of OTC.
It would also be worthwhile for the authors to indicate when to suspect this clinical entity and what laboratory parameters to request. Often, in most hospitals, an aminogram is not immediately available, and results are obtained several days later. In the case series provided, only 30% were able to obtain information on some of the amino acids and/or metabolites involved in this pathology. The simple question would be: should a OTC deficiency be suspected in the presence of severe hyperammonemia? In these cases, even if the disease is not confirmed, is it necessary to reduce protein intake and supplement with glucose infusions and vitamins and minerals? Please describe the actions to be taken in more detail.
Minor points:
Please also provide temperature measurements in degrees Celsius.
Point 2.1. Role of OTC Dysfunction in Fasting Hypoglycemia
“It is interesting to note that reactive hypoglycemia has been reported in post-gastric bypass patients, often attributed to nesidioblastosis of pancreatic islet cells with post-prandial hyperinsulinemia.”
This theory is practically ruled out or is rare. It is better to simply describe that “postprandial hypoglycemia is common in patients undergoing gastric bypass” (although the pathogenesis of hypoglycemia in these cases is unrelated to dumping syndrome or postprandial hypoglycemia, as the authors later confirmed).
Author Response
Thanks for your review of this manuscript. Here are some responses to your comments.
- You suggested a mention as to when to suspect this clinical entity. Accordingly, we added a few sentences in the manuscript addressing this excellent question (lines 355 to 359).
- We appreciate your comments about the possible role of the microbiota, but we had no data whatsoever regrading this issue, and hence elected not to include this issue in light of the absence of any data.
- As requested, we converted the Fahrenheit to Celsius in the case report.
- Also, we revised the section on the hypoglycemia issue, and indeed deleted a figure, and hence reduced the number of figures to 7 (from 8). See the revised discussion about the occurrence of hypoglycemia in our patients.
Reviewer 2 Report
Comments and Suggestions for Authors
In this systematic review, the authors presented a case of GaBHA syndrome. The hypothesis contends that arginine deficiency produces a functional deficiency of the ornithine transcarbamylase (OTC) enzyme, leading to severe non-cirrhotic hyperammonemia. The work is compiled well in this review. I have the following comments about this review.
- How did the author identify the various metabolites in the different studies? Compile that information in a separate section.
- Did the author report and find any fluxomics studies using stable isotope tracers for pathway analysis in metabolism?
- The resolution of Figures 4a and 7 is not appropriate. Make the figures with better resolution.
- The conclusion is not critical. Rewrite it to reflect the critical analysis of the authors.
- The abstract is not structured properly. The introduction and objective are not clear in the abstract.
Author Response
Thanks for your review of this manuscript. Here are some responses to your comments.
- The first question was identification of the various metabolites. I think this refers to the organic acids, and this was identified in our patients and those in the literature as reported by the organic acid profiles. This is clarified in the manuscript.
- Did the authors find any fluxomics studies using isotope tracers for pathway analysis? No, we did not, and hence this was not mentioned.
- The resolution of figures 4 and 7 are poor. We fully agree and have markedly improved this in the new version. Please also notice we reduced the figures to a total of 7 (from 8).
- The conclusion is not critical. We agree and edited this section accordingly.
- The abstract is not structured properly. This was corrected accordingly. See the current abstract.